# Prospective Analysis of Prevalence, Trajectories of Change, and Correlates of Cannabis Misuse in Older Adolescents from Coastal Touristic Regions in Croatia

**DOI:** 10.3390/ijerph16162924

**Published:** 2019-08-15

**Authors:** Lejla Obradovic Salcin, Vesna Miljanovic Damjanovic, Anamarija Jurcev Savicevic, Divo Ban, Natasa Zenic

**Affiliations:** 1Clinic for Physical Medicine and Rehabilitation, University Hospital Mostar, Mostar 88000, Bosnia and Herzegovina; 2Faculty of Health Sciences, University of Mostar, Mostar 88000, Bosnia and Herzegovina; 3Teaching Institute of Public Health of Split Dalmatian County, Split 21000, Croatia; 4Faculty of Kinesiology, University of Split, Split 21000, Croatia; 5University of Dubrovnik, Dubrovnik 22000, Croatia

**Keywords:** marijuana, puberty, sport participation, tourism, environment

## Abstract

The prevalence of illicit drug misuse, including cannabis, in Croatian touristic regions is alarming. This study aimed to identify the prevalence of cannabis consumption (CC), to identify associations between sociodemographic and sport factors and CC, and to evaluate the predictors of CC initiation in adolescents residing in touristic regions. This study enrolled 644 adolescents from two touristic regions in Croatia (Split-Dalmatia and Dubrovnik-Neretva County) who were tested at baseline (16 years of age) and follow-up (18 years of age). The study instrument consisted of questions focused on predictors (age, gender (male, female), place of residence (urban or rural environment), familial social status, and different sport-related factors) and CC outcome. The results indicated a high prevalence of cannabis consumption (>30% of adolescents consumed cannabis), with a higher prevalence in males, and adolescents from rural communities. The prevalence of CC increased by 10% during the study period, with no significant differences between genders in trajectories of changes. Quitting sports was a risk factor for CC at baseline and follow-up. Better sport competitive results (odds ratio (OR): 0.80, 95% confidence interval (CI): 0.65–0.96) and familial social status (socioeconomic status: OR: 0.66, 95% CI: 0.39–0.91; maternal education: OR: 0.65, 95% CI: 0.48–0.88) were associated with lower likelihood of CC at baseline. The adolescents who reported better sport competitive results were at increased risk for initiation of CC during the course of the study (OR: 1.40, 95% CI: 1.03–2.01). The protective effects of sports at baseline were most likely related to various factors that prevent the consumption of substances in youth athletes (i.e., commitment to results, adult supervision); with the end of active participation, adolescent athletes are at high risk for CC initiation.

## 1. Introduction

Croatia is a country located on the coast of the Adriatic Sea at the Balkan Peninsula. After the devastating wars in the early 1990s, the country experienced exponential growth in the touristic sector and became one of the most popular touristic destinations in the world [1,2]. While it is known that tourism has numerous positive effects on society (increased employment, greater income for the local economy, and preservation and improvement of local services), the rapid rise of the touristic sector also results in some negative consequences (i.e., over-tourism, a negative impact on the environment, traffic congestion, and commercialization) [3]. From the public health perspective, one of the most important negative consequences of rapid tourism development is the increased prevalence of substance misuse in touristic regions [4,5,6]. For example, Krizman et al. recently performed wastewater analysis and investigated spatial and temporal consumption patterns of selected illicit drugs and opioids in six Croatian cities [5]. Among other conclusions, the study highlighted a significant increase in illicit drug prevalence in the coastal touristic region, which was connected to a specific touristic lifestyle [5]. This is directly supported by recent reports of the Croatian Institute of Public Health who emphasized the highest rates of medically treated drug addicts specifically for coastal–touristic regions [7].

Cannabis (marijuana) was the most cultivated, trafficked, and used illicit drug in 2016 globally, with 192 million people using it at least once in the last 12 months, both among the general population and among young people [8]. The level of harmful marijuana consequences depends on a variety of factors, such as dose, potency, and cumulative exposure, and adolescents are known to be a particularly vulnerable group [9]. Specifically, a recent study suggested structural brain and cognitive effects after extremely low levels of cannabis use in adolescence (such as just one or two instances) [10]. Negative long-term effects include cognitive decrements and neural changes, poorer verbal learning, and reduced memory [11,12]. Apart from direct negative health effects, cannabis usage is connected to an increased risk of driving accidents, and even acute cannabis consumption nearly doubles the risk of a collision due to impaired motor functions [13,14].

The pathway from experimental to harmful marijuana use among adolescents is mostly influenced by factors that are often out of their control. There are a variety of factors, including personal factors (such as genetic susceptibilities, mental health and personality traits, neurological development, and stress reactivity), as well as micro-level (parental, family, school, and peer influences) and macro-level (socioeconomic and physical environment) factors, which may influence the vulnerability of adolescents to cannabis misuse [8]. According to the “social–ecological model”, in order to understand human development and lifelong changes (including behavioral changes), the entire ecological system in which growth and development occur should be taken into account [15]. Indeed, there is no doubt that the immediate physical and social environment, as well as interactions among the systems within the environment, affects one’s life and behavior. In the period of adolescence, this is particularly evident because the young people function in various environments, constantly trying to position themselves in the most comfortable environment. Therefore, taking into account influence of the specific environment in which the adolescents develop, such as residing in touristic regions, is of upmost importance even in the domain of public health [16,17]. 

One of the social–ecological factors which deserves attention as being potentially related to consumption of illicit drugs, including cannabis, is participation in sport. Wichstrøm and Wichstrøm identified several theoretically protective aspects of sport with regard to risk of cannabis misuse: (i) age segregation (i.e., age segregation is common in sports, which consequently decreases the possibility of bonding with older adolescents and consequently reduces the risk of drug use), (ii) time occupation (i.e., sports training and competitions take time and, therefore, there is less time for activities associated with consumption of psychoactive substances, including cannabis), (iii) adult supervision (i.e., adult coaches are regularly involved, which may limit problem behavior), and (iv) orientation toward success (i.e., consumption of illicit drugs reduces the physical capacities and, therefore, alters the sport results and achievement). On the other hand, authors also recognized sport as a social activity that may present a certain risk of a higher likelihood of substance misuse, including cannabis consumption. Putting it altogether, many but not all sport characteristics may reduce the risk of cannabis use [18]. Supportively, studies which investigated the associations between sport factors and illicit drug misuse reported inconsistent findings [19,20,21].

In recent years, there was growing interest in factors associated with substance misuse among adolescents in the territory of Croatia, mostly because reports showed alarming figures of substance misuse not only in the country but also in the whole region [20,22,23]. Specifically, most dramatic are the figures on cigarette smoking and alcohol drinking that put Croatian adolescents among the top 10% of the most vulnerable adolescent groups in Europe [24]. As a result, recent studies systematically investigated the factors of influence on smoking and drinking [25,26]. However, apart from studies that included data from Croatia in international reports (i.e., European Survey Project on Alcohol and Other Drugs (ESPAD), there is an evident lack of studies that were specifically focused on cannabis, which is the most frequent illicit drug consumed in Croatia [27,28,29]. Additionally, to the best of our knowledge, no study so far specifically focused on the consumption of drugs and correlates of this behavior in Croatian touristic regions.

Therefore, the aims of this study were to prospectively evaluate the prevalence and trends in cannabis misuse, as well as correlations between sociodemographic and sport factors (predictors) with cannabis misuse and cannabis initiation, among 16- to 18-year-old Croatian adolescents residing in the coastal (touristic) part of the country. Herein, we were specifically interested in sociodemographic and sport factors as possible predictors of cannabis misuse, mainly on the basis of conclusions from recent studies that showed promising results while prospectively investigating similar correlates of cigarette smoking and alcohol drinking [25,26]. Initially, we hypothesized the following: (i) there is a high prevalence of cannabis consumption among adolescents from Croatian touristic regions, (ii) there is a protective effect of sport participation against cannabis consumption, and (iii) there is a higher likelihood of cannabis misuse in adolescents of lower socioeconomic status.

## 2. Methods

### 2.1. Design and Setting

In this prospective observational study, a multistage sampling procedure was applied (Figure 1). We observed a cohort of 644 adolescents who were aged 16.4 years (±5 months) at study baseline. The study consisted of baseline testing (October 2014) and follow-up (May 2016) in two Croatian counties, Split-Dalmatia and Dubrovnik-Neretva County (Figure 1). These two neighboring counties were selected on the basis of their specific position on the coast of the Adriatic Sea, the southern part of Croatia, and their strong orientation toward touristic economy. While both counties are geographically very narrow and positioned over the Croatian coast (please see Figure 1), nearly the whole territory of these counties is strongly associated with the touristic sector, which makes them particularly convenient for research of such kind.

### 2.2. Study Population and Inclusion/Exclusion Criteria

The aim of this investigation was to observe the period of life between 16 and 18 years of age. In general, this period of adolescence corresponds with the last two years of high-school education (i.e., the educational system in Croatia consists of eight years of primary school (starting from 6–7 years of age), followed by 3–4 years (depending on program) of high-school education). Basically, this study was designed as two-step testing: in the beginning of the third year of high school (baseline; mostly sixteen-year-olds pupils), and after approximately 20 months (follow-up; age 18, on average). The total theoretical sample of participants (4494 third-year pupils of all 121 high schools in two participating counties) was divided into three clusters based on the number of pupils in each school. In each cluster, we randomly selected 30% of the schools (39 schools). Then, for those schools who have lessons in two shifts, we randomly selected one school shift. At the last stage of sampling, we selected only those third-year classes that had a four-year high school program (note that some high school programs (so-called “professional educations”) are organized across three-year study programs), which resulted in a crude study sample of 37 classes and 856 pupils.

The study was approved by the Ethics Committee of the University of Split, Faculty of Kinesiology (EBO: 2181-205-05-02-05-14-005). Participation in this study was voluntary and was offered to all pupils in the selected cohort. Prior to study, parents were informed about the purpose, risks and benefits of testing, and were asked to sign the informed consent for participation of their child/children. We excluded all pupils who were not present in the school on one of the testing days, since the testing was planned to be performed only once for baseline and once for follow-up. This built a final study sample of 644 participants (298 females) (Figure 1).

After the second wave of testing, we calculated an analysis of attrition bias. Specifically, the χ^2^ test evidenced no significant difference in baseline cannabis consumption between those adolescents who were tested both at baseline and follow-up, and those who dropped out (chi-square (χ^2^): 1.13, *p* > 0.05). Furthermore, significantly more males than females dropped out from the study (χ^2^ = 15.13, *p* < 0.01), which may be attributable to higher absence from school among boys [30]. Finally, the intracluster correlation (IC) for the baseline cannabis consumption (with the schools as clusters) showed appropriate (i.e., <0.10) within-school variance (IC = 0.06 and 0.05 for baseline and follow-up) [31,32].

### 2.3. Instrument

In this study, we applied the Questionnaire of Substance Use (QSU), which was previously validated and found to be applicable in the local language [30]. The study instrument consisted of questions focused on predictors (age, gender (male, female), place of residence (later grouped in urban and rural environment), familial social status, and sport factors), and the outcomes (consumption of cannabis at baseline and follow-up, and initiation of cannabis consumption).

The familial social status was determined by three questions: one asking for self-determined socioeconomic status (SES; under average, average, and above average), one question on maternal education, and one question for paternal education (three possible answers for both maternal and paternal education: elementary school, high school, and college/university degree). Although these three variables theoretically determine different social facets (i.e., economics, educational status), similar indices were used previously in order to identify construct of social status [33].

Although many studies examined how sport participation is associated with substance use in adolescence, only a few studies explored this association separately for individual and team sports [26,34,35] as was done in this study. Special attention was paid to the overall lifetime level of experience in sports since a number of studies concluded that substance use may vary according to current and not overall sport activities [20,36]. Therefore, the pupils were asked not only whether they participated in sports but also whether he/she quit sports at some time. The questions about sport factors included (i) involvement in competitive individual sports and (ii) involvement in competitive team sports (each of them reported as never involved, quit, or currently involved), (iii) highest achieved competitive sport result (never involved/competed, competed locally, competed nationally/internationally), and (iv) duration of sport involvement (never participated, participated for less than one year, participated for 2–5 years, participated for more than 5 years).

Possible responses to the question assessing cannabis consumption included “never”, “tried only once”, “2–3 times”, “more than three times so far”. Participants were later grouped as “non-consumers” (who responded “never tried”), and “consumers” (remaining responses). Additionally, those adolescents who were cannabis consumers at follow-up and non-consumers at baseline (e.g., who responded “never tried” at study baseline, and gave a different response at follow-up) were counted as those who initiated cannabis consumption during the course of the study.

It took 15 min on average to complete the questionnaire; the questionnaire was completed by pupils during a normal class period and put in a closed box. No identifying data were collected; however, in order to match the results from the two testing waves, participants were instructed to use a self-selected and easy to remember username (such as the last three digits of their e-mail password).

### 2.4. Statistics

The chi-square test (χ^2^) was calculated to identify differences between genders and urban/rural environment in prevalence of cannabis misuse. Three sets of logistic regressions were calculated to identify the correlates of cannabis consumption/initiation. Specifically, we calculated logistic regression between predictors and cannabis consumption at (i) baseline and (ii) follow up, and these analyses included all participants. Finally, predictors were correlated with (iii) cannabis initiation during the course of the study; for this calculation, we included only those adolescents who were non-consumers at baseline. The model fit was checked by Hosmer–Lemeshow test (with significant χ^2^ indicating inappropriate model fit). Since preliminary statistical procedures showed significant association between (i) gender and cannabis misuse, and (ii) urban/rural environment and cannabis misuse (see later text for details on differences between genders and urban/rural environment), logistic regressions were adjusted for gender and urban/rural environment.

Statistica ver. 13.5 (Tibco Inc., Palo Alto, Ca) was used for all analyses, and a significance level of *p* < 0.05 was applied.

## 3. Results

The descriptive statistics for studied variables at baseline and follow-up are presented in the Appendix A.

The prevalence of cannabis misuse was higher in males than in females at both testing waves (χ^2^: 13.68 and 13.92, *p* < 0.01, for baseline and follow-up, respectively). The prevalence of cannabis misuse increased from 28% at study baseline to 37% at follow-up (χ^2^: 12.59, *p* < 0.01), with a similar increase across genders. The prevalence of cannabis consumption was higher in rural than in urban communities at study baseline (χ^2^: 5.16, *p* < 0.05), with no significant difference between communities at follow-up (χ^2^: 1.13, *p* = 0.29) (Figure 2).

The results of logistic regression for the binominal criterion “cannabis misuse at study baseline” are presented in Table 1. The higher likelihood for cannabis misuse at baseline was evidenced for older adolescents (odds ratio (OR): 1.53, 95% confidence interval (CI): 1.01–2.31), and those who quit sports (OR: 2.78 and 1.21, 95% CI: 1.71–4.52 and 1.02–1.87, for individual and team sports, respectively). Meanwhile, those adolescents who perceived their familial socioeconomic status as higher (OR: 0.66, 95% CI: 0.39–0.91), whose mothers were better educated (OR: 0.65, 95% CI: 0.48–0.88), and those who achieved better sport competitive results (OR: 0.80, 95% CI: 0.65–0.96) were less likely to consume cannabis at baseline (Hosmer–Lemeshow test; χ^2^: 8.21, *p* = 0.41).

The correlates of cannabis misuse at follow-up are presented in Table 2. The Hosmer–Lemeshow test indicated inappropriate model fit (χ^2^: 41.14, *p* < 0.01), with higher likelihood for cannabis misuse at the end of the fourth year of high school for adolescents who reported that they quit individual sports (OR: 1.75, 95% CI: 1.13–2.73).

When logistic regressions were calculated for binomial criterion “initiation of cannabis misuse”, (χ^2^: 13.11, *p* > 0.05), the higher likelihood for cannabis initiation was found among those adolescents who reported better sport competitive results (achievement) at study baseline (OR: 1.40, 95% CI: 1.03–2.01) (Table 3).

## 4. Discussion

There are several important findings of this study with regard to the study aims. Firstly, the prevalence of cannabis consumption in adolescents from Croatian touristic regions was high at baseline and increased during the course of the study by approximately 10% with no significant difference in trajectory of increase according to gender or rural/urban residence. At baseline, the increased risk for cannabis consumption was found for older children, children of lower familial social status, and those who quit sports. Adolescents who achieved sport success were at lower risk for cannabis misuse at baseline. Adolescents who were more successful in sports were more vulnerable to the initiation of cannabis misuse during the course of the study. Collectively, we may confirm the first (high prevalence of cannabis consumption among adolescents from Croatian touristic regions) and third hypotheses (higher likelihood of cannabis misuse in adolescents of lower socioeconomic status). Meanwhile, the second hypothesis of the study (protective effect of sport participation against cannabis consumption) can be partially accepted.

### 4.1. Prevalence and Trajectory of Cannabis Consumption in Croatian Adolescents

According to the presented results, we can say that the prevalence of cannabis consumption in our sample (approximately 30% users; 95% CI: 26–33%) was higher than the average value reported for the whole territory of Croatia (22% of users) [37]. However, these differences are actually expected. Specifically, as already specified in the introduction, Krizman et al. performed a very recent environmental study in which a wastewater-based assessment identified regional patterns of illicit drugs in Croatia and highlighted clear differences between Croatian regions in the consumption of all kinds of illicit drugs. Generally, consumption was evidently higher in coastal touristic regions, which was explained by the “lifestyle in coastal tourist centers” [5].

The high prevalence of cannabis consumption in our sample is in accordance with public health guidelines obtained in a recent ESPAD report. In short, from 1995–2003, an almost triple growth of illicit drug use among youth (age 15–16 years) was noted in Croatia, with a slight decrease in 2007 and 2011, but the values increased again in 2015 to the 2003 level [38]. The research showed concerning results in general, especially in comparison with the average of European countries. Approximately 22% of youths in Croatia vs. 18% in Europe used marijuana, which was the most prevalent drug, at least once during their lifetime.

Our results are comparable to those presented for adolescents from Bosnia and Herzegovina, a Croatian border country [30,39]. Specifically, the prevalence of cannabis consumption is evidently much higher in Croatia (35% cannabis consumers) than in Bosnian and Herzegovinian adolescents (less than 10% cannabis consumers) [30,39]. This striking difference between neighboring countries is probably connected to differences in socioeconomic status between the countries (i.e., Croatian gross national income (GNI) is almost twice that of Bosnia and Herzegovina at 12,000 and 6000 United States dollars (USD) per capita, respectively), while the “true difference” is almost certainly even larger if we take into account that we observed Croatian cantons with the highest GNI in the country (due to high touristic income). Secondly, in Croatia (and especially in touristic regions), illicit drugs are common and available [37]. We must say that this problem was predicted by responsible authorities and academicians even back in the late 1990s [28]; however, it seems that preventive public health preventive policies were not effective. Additionally, we must not ignore the possible negative consequences of rapid development in the touristic sector [4,5,40].

The previous discussion of the prevalence of cannabis use is directly supported by the significant increase in consumption of this illicit drug during the studied period (an increase of approximately 10% over two years). In general, the increase is expected, especially knowing that similar trends were already reported for other countries in same age group. For example, in a previously cited study that investigated the problem in Bosnia and Herzegovina, the authors reported an increase of 3–4% for illicit drug consumption over a two-year period [35,39]. Knowing the difference in prevalence of consumption between Croatia and Bosnia and Herzegovina (30% and 10% consumers, respectively; please see previous text for details), the relative changes in trajectories in the two countries are actually similar.

Although females were less likely to be cannabis consumers than males, the trajectories of increase in cannabis misuse are similar across genders (35% to 45% and 21% to 30% for males and females, respectively). One of the probable reasons for a similar increase in prevalence between genders could be found in the overall perception of males and females about the obtainability of cannabis in Croatia. In short, in a recent report 41.7% of females and 41.4% of males were of the opinion that “cannabis is easily obtainable”, and only 14.4% of females and 14.8% of males shared the opinion that “it is impossible to obtain cannabis” [37].

### 4.2. Correlates of Cannabis Consumption in Croatian Adolescents

Socioeconomic status is frequently studied as a potential factor of influence on illicit drug consumption, including cannabis consumption in youth [33,41,42]. Although not all studies confirmed a statistically significant association between SES and consumption of cannabis in adolescence, the previously published results provide general support for the existence of an association between social disadvantage (including low SES) and cannabis consumption [29,43]. Since we found a higher risk for cannabis misuse in adolescents with lower SES, we may say that our results are in agreement with studies that reported similar correlations in New Zealand, United States of America (USA), and Sweden [44,45,46]. The explanation of such relationships is certainly interesting and should be explained in more detail, but this would be beyond the scope of this investigation. However, we may support conclusions of previous studies in which authors concluded that cannabis consumption should be observed as a specific sequela of low SES and childhood disadvantage [33].

Studies frequently observed the associations between sport participation in adolescence and illicit drug misuse (including cannabis consumption), but the relationships that exist between sport participation and cannabis consumption are not clear. The study by Torstveit and al, which included more than 3200 junior-high and high-school students in southern Norway, found that organized sport participation was consistently associated with decreased odds of unhealthy lifestyle habits, including cannabis misuse [47]. Brellenthin et al. reviewed the epidemiological literature to describe the associations of physical activity and substance misuse across the lifespan and concluded that physical activity is negatively associated with drug use [48]. With regard to cannabis consumption, some studies reported no relationship and some reported positive associations (i.e., protective effect of sport); however, in other studies, authors highlighted negative associations between sports participation and marijuana use [19,49,50]. Therefore, the somewhat contradictory findings from our study that adolescents who achieved high sport success were at lower risk for cannabis misuse at study baseline but were at higher risk to initiate cannabis consumption during the period of the study are not uncommon.

The association between sport achievement and cannabis misuse at study baseline can be interpreted from two perspectives: one in which sport commitment (and consequent success in sport) is a factor that prevents cannabis misuse and another in which cannabis misuse is related to poorer sport achievement. While both mechanisms of influence are actually possible, we briefly discuss the idea and theoretical background for each one. One of the most important motives for sport participation is motivation to succeed [51]. Success in sports is unachievable without commitment to all facets of sport participation, including training and competition, proper eating, rest, and a general healthy lifestyle. Illicit drug consumption, including cannabis misuse, alters physical capacities via numerous mechanisms that are connected but not limited to diminished cognitive functioning, increased heart rate, and reduced capacity of the blood to carry oxygen [52]. Notably, those adolescents who consume cannabis are less likely to be properly physically fit to achieve highly competitive results in sport.

On the other hand, it is possible that cannabis misuse should be observed as a factor that is actually a result of poor competitive results. Indeed, recent studies that examined sport factors as correlates of smoking and alcohol drinking in some circumstances actually confirmed that poor sport success is “the cause” of substance misuse throughout a specific social context. Briefly, sport is “social activity”, while substance misuse occurs in a “social environment”. Therefore, athletic adolescents who are not successful are more likely to consume substances (including cannabis) because of their tendency toward “social activities”, while the lack of competitive achievement in sports actually prevents them from observing substance misuse as a factor that negatively influences their achievement in sport. One can argue that it is possible that association is actually a result of a higher prevalence of cannabis consumption in athletes who never participated in sport, but this was not the case since logistic regression clearly indicated that the highest risk was evident in those adolescents who participated in sports but did not achieve high success. 

In explaining findings of higher risk for cannabis initiation in the studied period for those who reported better competitive achievement in sport at study baseline, some contextual factors are important. Specifically, in Croatia, the age of 17 years is known to be a critical time point for participation in competitive sports. Specifically, at this age, the largest sport participation drop-out rate occurs because professional involvement begins at the age of 18 years in most sports. Therefore, only those adolescents whose sport capacities fulfill the needs of senior-level training and competition progress in sport participation [53]. It is also important to note that former athletes probably have a certain tendency toward meeting new friends (i.e., those out of sports), which consequently “increases the risk” of being involved in sociocultural circumstances where cannabis is consumed, resulting in possible initiation of cannabis usage in those who reported sport participation at the baseline testing.

It is important to note that the previous mechanism of the theoretical influence of sport factors on cannabis initiation is particularly possible for adolescents in touristic regions since, in this period of life, local youths are intensely engaged in different branches of the touristic sector [54]. Consequently, this results both in (i) the tendency to quit sports and (ii) a higher availability of money. For those adolescents who quit sports, all previously mentioned “protective” mechanisms of sport participation (i.e., age segregation, time occupation, adult supervision, and orientation toward success) suddenly disappear [18]. Since males generally start to work in tourism earlier than females, this is one of the possible reasons for the higher prevalence of cannabis consumption at study baseline among males.

Although further investigations are needed to support previous considerations about negative influence of sport factors on cannabis initiation, previous discussion can be indirectly supported by results of some previous studies where authors investigated correlated sport factors with consumption of illicit drugs among adolescents. For example, a very recent longitudinal study of almost 2000 Canadian adolescent females who participated in team sports revealed an increase in the odds of becoming more frequent cannabis users [50]. Additionally, our results about increased risk for cannabis initiation in athletic adolescents is consistent with some cross-sectional and prospective reports in the territory of southeastern Europe [20,35]. Collectively, results highlight the possibility that, under some specific circumstances, sport participation should be observed as being negatively related to consumption of illicit drugs (including cannabis).

### 4.3. Limitations and Strengths

This study was based on self-report, and this is certainly the most important limitation of the investigation. Consequently, the participants may lean toward socially desirable answers. Additionally, the study was exclusively based on quantitative data; therefore, some important qualitative factors that would provide more clear insight into the studied relationships are missing. Another important limitation of the study is related to recall bias. In short, it is possible that participants did not properly remember some of the facts they were asked about (i.e., number of times they consumed cannabis); however, since we observed cannabis consumption on a nominal scale (i.e., consumed vs. non-consumed) we believe that this problem did not influence our findings significantly. Finally, the study was focused on only two specific counties from Croatia and, therefore, the generalizability of the findings is limited. However, we were particularly interested in the studied regions because of the evidence that touristic regions of the country are especially vulnerable with regard to the consumption of illicit drugs.

This is one of the first studies that prospectively investigated the problem of cannabis use in touristic regions and is probably the first to systematically investigate factors associated with cannabis misuse and initiation in this part of Europe. Furthermore, sport factors were extensively studied, which provided us with the opportunity to discuss the results in detail. Finally, the previously validated measurement instruments and the high retention rate are important strengths of the investigation.

## 5. Conclusions

Although this study was based on self-report, the results indicate a high prevalence of cannabis consumption in adolescents from touristic regions in Croatia. In the last 10 years, both studied counties experienced rapid development of the tourism sector, which is still seemingly rising. Some positive impacts are undeniable (i.e., significant contribution to financial income and overall economic benefits); however, some of the impacts have negative influence and almost certainly include the increase in availability and consequent increase in consumption of illicit drugs.

Trajectories of changes in cannabis consumption between the ages of 16 and 18 years showed similar increases of approximately 10% for both males and females. On the other hand, males were more vulnerable to cannabis consumption both at study baseline and follow-up. Therefore, it seems that the background for the higher risk of cannabis consumption in males should be investigated in younger age groups.

The adolescents with lower SES were found to be at higher risk of cannabis consumption than their peers with better finances. However, it is questionable whether participants objectively identified their SES status or whether their evaluation was influenced by comparison to their closest friends. Therefore, future studies should use additional measures of SES to identify the plausibility of the reported results.

While sport participation and success were protective against cannabis misuse at study baseline (16 years of age), the adolescents who were successful in sports were more vulnerable to initiate cannabis misuse during the course of the study (between 16 and 18 years of age). The protective effects of sports are almost certainly related to various factors that prevent the consumption of substances in youth athletes (i.e., commitment to results, adult supervision). On the other hand, in this period of life, many children leave the sport and become vulnerable to different negative social influences, including those connected to cannabis consumption.

This research is in line with proposed activities to reduce harmful cannabis use disorders that are fully preventable. There is an urgent need to support and carry out drug-related research among risk groups, such as adolescents, and to investigate the social and public health impact of cannabis misuse. Therefore, we hope that this study may help to create and implement evidence-based programs with the highest priority given to intervention targeting adolescents.

## Figures and Tables

**Figure 1 ijerph-16-02924-f001:**
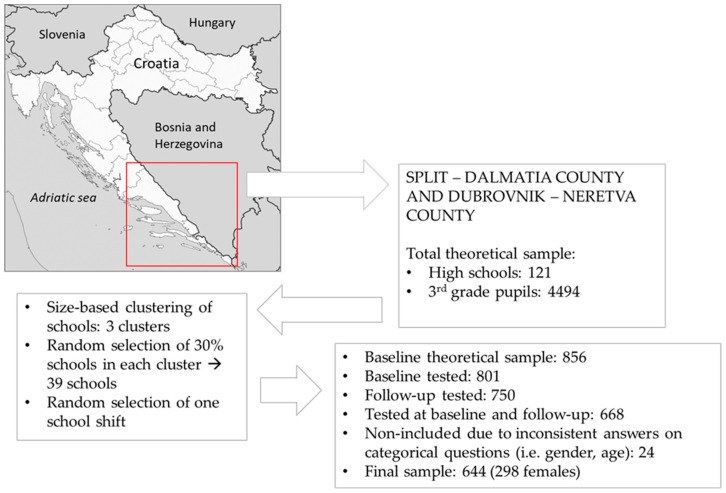
Location of the study, testing sequences, and drop-out rates of the study sample.

**Figure 2 ijerph-16-02924-f002:**
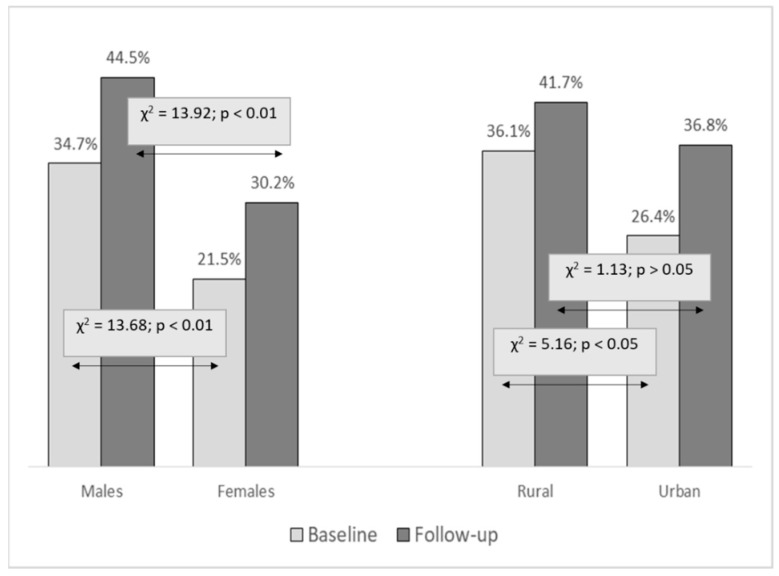
Prevalence of cannabis misuse at baseline and follow-up for genders and urban/rural environment with chi-square test differences (χ^2^) between genders and rural/urban environments.

**Table 1 ijerph-16-02924-t001:** Correlates of cannabis misuse at baseline (adjusted for gender and urban/rural environment). OR—odds ratio; CI—confidence interval.

Baseline (*n* = 644)	Logistic Regression	
	OR	95% CI
Age ^cont^	1.53	1.01–2.31
Participation in individual sport		
Yes	1.07	0.59–1.95
Quit	2.78	1.71–4.52
No	Reference	
Participation in team sport		
Yes	1.07	0.53–2.17
Quit	1.21	1.02–1.87
No	Reference	
Experience in sport ^cont^	0.85	0.63–1.14
Sport competitive result ^cont^	0.8	0.65–0.96
Socioeconomic status ^cont^	0.66	0.39–0.91
Paternal education ^cont^	1.28	0.98–1.67
Maternal education ^cont^	0.65	0.48–0.88
Hosmer–Lemeshow fit test		
χ^2^	8.21	
*p*	0.41	

Legend: ^cont^ indicates variables observed as continuous for the purpose of the logistic regression calculation.

**Table 2 ijerph-16-02924-t002:** Correlates of cannabis misuse at follow-up (adjusted for gender and urban/rural environment).

Follow-Up (*n* = 644)	Logistic Regression	
	OR	95% CI
Age ^cont^	0.99	0.69–1.42
Participation in individual sport		
Yes	1.6	0.97–2.67
Quit	1.75	1.13–2.73
No	Reference	
Participation in team sport		
Yes	1.22	0.67–2.23
Quit	0.83	0.53–1.32
No	Reference	
Experience in sport ^cont^	0.73	0.56–0.95
Sport competitive result ^cont^	1.32	0.93–1.60
Socioeconomic status ^cont^	1.46	0.93–2.28
Paternal education ^cont^	1.12	0.88–1.43
Maternal education ^cont^	0.97	0.75–1.26
Hosmer–Lemeshow fit test		
χ^2^	41.14	
*p*	0.01	

Legend: ^cont^ indicates variables observed as continuous for the purpose of the logistic regression calculation.

**Table 3 ijerph-16-02924-t003:** Correlates of cannabis misuse initiation during the course of the study (adjusted for gender and urban/rural environment).

Initiation (*n* = 99)	Logistic Regression	
	OR	95% CI
Age ^cont^	0.79	0.47–1.33
Participation in individual sport		
Yes	1.38	0.73–2.63
Quit	1.11	0.59–2.09
No	Reference	
Participation in team sport		
Yes	2.05	0.88–4.72
Quit	1.06	0.53–2.09
No	Reference	
Experience in sport ^cont^	0.72	0.49–1.05
Sport competitive result ^cont^	1.4	1.03–2.01
Socioeconomic status ^cont^	1.22	0.67–2.21
Paternal education ^cont^	0.98	0.69–1.39
Maternal education ^cont^	1.29	0.92–1.84
Hosmer–Lemeshow fit test		
χ^2^	13.11	
*p*	0.09	

Legend: ^cont^ indicates variables observed as continuous for the purpose of the logistic regression calculation.

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
