# Peer review of "Prospective Analysis of Prevalence, Trajectories of Change, and Correlates of Cannabis Misuse in Older Adolescents from Coastal Touristic Regions in Croatia"

_ijerph, 2019, doi:10.3390/ijerph16162924_

Round 1
Reviewer 1 Report
Good overall, several points of clarification requested:
One of the factors considered is “cannabis consumption (CC). Is this considered to be an increase if somebody uses cannabis once after not using at baseline? This was unclear to me, as it discussed about overall cannabis consumption. For the comparison with bordering countries, the rate of cannabis use is compared with overall illicit substance use. This isn’t a clear comparison, are statistics available for cannabis use in bordering countries for a clear comparison? In the discussion section, I’d consider recall bias as a potential limiting factor. As somebody from the United States/ internationally located, a brief overview of the Croatian schooling system would be helpful. Particularly when considering the impact of this specific age demographic that was selected.
Author Response
First of all, we must thank you for your comments and suggestions. WE tried to follow it specifically and amended the manuscript. Please see in the following text for RESPONSES.
Staying at your disposal.
REVIEWER 1
One of the factors considered is “cannabis consumption (CC). Is this considered to be an increase if somebody uses cannabis once after not using at baseline? This was unclear to me, as it discussed about overall cannabis consumption.
RESPONSE: Yes, for the purpose of this study as “those who initiated with cannabis consumption” we treated all participants who reported “never tried” at study baseline, and any other response at follow-up. This is more specifically explained in the text now, and reads: “Additionally, those adolescents who were cannabis consumers at follow-up and nonconsumers at baseline (e.g. who responded “never tried” at study baseline, and gave different response at follow-up) were counted as those who initiated cannabis consumption during the course of the study.” (Please see 4th paragraph of the subsection 2.3. Instrument)
For the comparison with bordering countries, the rate of cannabis use is compared with overall illicit substance use. This isn’t a clear comparison, are statistics available for cannabis use in bordering countries for a clear comparison?
RESPONSE: The text is amended according to your suggestion, and we compared our data with cannabis consumption in boarder countries. Text now reads: “Our results are comparable to those presented for adolescents from Bosnia and Herzegovina, a Croatian border country [24,31]. Specifically, the prevalence of cannabis consumption is evidently much higher in Croatia (35% cannabis consumers) than in Bosnian and Herzegovinian adolescents (less than 10% cannabis consumers) [24,31].” (Please see 3rd paragraph of the subsection 4.1)
In the discussion section, I’d consider recall bias as a potential limiting factor.
RESPONSE: Thank you for noticing it. We included the problem of recall bias in Study limitations subsection. Text reads: Another important limitation of the study is related to recall bias. In short, it is possible that participants did not properly remember some of the facts they were asked about (i.e. number of times they consumed cannabis), but since we observed cannabis consumption on nominal scale (i.e. consumed vs. non-consumed) we believe that this problem didn't influence our findings significantly. (Please see highlighted text within Limitations and strengths of the study).
As somebody from the United States/ internationally located, a brief overview of the Croatian schooling system would be helpful. Particularly when considering the impact of this specific age demographic that was selected.
RESPONSE: Indeed, this issue was not properly explained in the original version. In this version the scholastic system in Croatia is briefly explained. Text reads: „In general, this period of adolescence corresponds with last two years of high-school education (i.e. educational system in Croatia consists of 8 years of primary school [starting from 6-7 years of age), followed by 3-4 years [depending on program] of high-school education).“ (Please see highlighted text in subsection Study population)
Reviewer 2 Report
This paper presents a true innovate approach in the chosen domain. Its originality and methodological excellence will simulate further research about this problematic.
1. On introduction, as a way to better understand the studies, the authors should adress an extended literature review of social ecological theory and methods.
2.On line 98, authors describe an observational study, they should name the type of sampling performed.
We do not have information about the number of classes in the study. We suppose that the authors used aggregated information fo all classes but perhaps it would be interesting to know the number of classes and the structural information desaggregated by those classes.
Author Response
Thank you for your review. Please see bellow how we responded to your comments and suggestions.
Staying at your disposal.
Authors
REVIEWER 2
This paper presents a true innovate approach in the chosen domain. Its originality and methodological excellence will simulate further research about this problematic.
RESPONSE: Thank you for recognizing the quality of our paper. We tried to improve it additionally while following your, and suggestions raised by other two Reviewers. Please see bellow how we amended the manuscript.
On introduction, as a way to better understand the studies, the authors should adress an extended literature review of social ecological theory and methods.
RESPONSE: Thank you for noticing it. In the revised version of the paper we shorthly overviewed the background of social ecological theory and its possible implications in public health. Text reads: “According to “social-ecological model” in order to understand human development and lifelong changes (including behavioral changes), the entire ecological system in which growth and development occur should be taken into account [15]. Indeed, there is no doubt that immediate physical and social environment as well as interactions among the systems within the environment affect one’s life and behavior. In the period of adolescence this is particularly evident because the young people function in various environments, constantly trying to position themselves in the most comfortable environment. Therefore, taking into account influence of specific environment in which the adolescents develop, such as residing in touristic regions, is of upmost importance even in the domain of public health [16,17]. “ (Please see 3rd paragraph of the Introduction).
2.On line 98, authors describe an observational study, they should name the type of sampling performed.
RESPONSE. Amended accordingly. Text reads: “In this prospective observational study, the multistage sampling procedure was applied (Figure 1).” (Please see first part of Design and setting subsection)
We do not have information about the number of classes in the study. We suppose that the authors used aggregated information fo all classes but perhaps it would be interesting to know the number of classes and the structural information desaggregated by those classes.
RESPONSE: Indeed, we did not specify the number of classes. It is specified now, and more details are provided with regard to its structure. Text reads: “Then, for those schools who have lessons in two shifts, we randomly selected one school shift. At the last stage of sampling, we selected only those 3rd year classes that had a 4-year high school program (note that some high school programs [so called “professional educations”] are organized across 3-year study programs), which resulted in a crude study sample of 33 classes and 792 pupils.” (Please see subsection 2.2. Study population – end of 1st paragraph) Thank you!
Reviewer 3 Report
This paper presents some interesting research that outlines some predictors of adolescent cannabis use in recently touristed regions in Croatia. I have several recommendations which I feel will improve the manuscript.
In the title, introduction, and abstract there is very little discussion about sport-factors, yet these feature prominently in the analysis, results, discussion and conclusion. It would be useful to reframe and balance the paper, so it is clear what the goals and theoretical underpinnings are of the research.
On p.134 – what value is considered “appropriate”?
It is not clear how the familial SES questions were brought together
Sex and urban rural status should be included in al models as covariates. These are important covariates and there is no theoretical or analytical reason not to. Thereby, only model 3 is needed and the coefficients for gender and urban/rural ought to be reported on.
The tables should include an n
On line 229 – it is not clear how it was determined that the trajectory was significantly increased
On lines 241-242 – confidence intervals should be given on the estimates to make better comparisons
Line 251- “unfortunately” is values based
Lines 166-167 are not necessary
On line 196 it seems that model 1 adjusts for no covariates but this is not the case in the tables
Some of the discussion is over-reaching the data- i.e., line 265, line 287-288
The final paragraph is vague.
Minor issues:
In the abstract on line 18, remove “(predictors)”
On line 27 – “was risk” should read “was a risk”
Line 33- the statement that the cannabis use found in the study is “alarming” is quite subjective. It would be more informative to compare the prevalence to other regions or remove this statement.
On line 129 – add who after adolescents
Author Response
Thank your for your Review and suggestions. We tried to follow your comments and specifically amended the manuscript. Please see bellow for responses.
Staying at your disposal.
Authors
REVIEWER 3
This paper presents some interesting research that outlines some predictors of adolescent cannabis use in recently touristed regions in Croatia. I have several recommendations which I feel will improve the manuscript.
RESPONSE: Thank you for your comments. We tried to follow it specifically, and amended the manuscript accordingly. Please see RESPONSE after each of your comment.
In the title, introduction, and abstract there is very little discussion about sport-factors, yet these feature prominently in the analysis, results, discussion and conclusion. It would be useful to reframe and balance the paper, so it is clear what the goals and theoretical underpinnings are of the research.
RESPONSE: Thank you for noticing it. Indeed, the first part of the manuscript lack information on sport factors as potential correlates of cannabis consumption. Therefore, in this version of the manuscript we added one paragraph of the Introduction where we explained potential effects of sport participation on misuse of drugs. Text reads: “One of the social-ecological factors which deserves attention as being potentially related to consumption of illicit drugs, including cannabis, is participation in sport. Wichstrøm and Wichstrøm identified several theoretically protective aspects of sport with regard to risk of cannabis misuse: (i) age segregation (i.e., age segregation is common in sports, which consequently decreases the possibility of bonding with older adolescents and consequently reduces the risk of drug use), (ii) time occupation (i.e., sports training and competitions take time, and therefore, there is less time for activities associated with consumption of psychoactive substances, including cannabis), (iii) adult supervision (i.e., adult coaches are regularly involved, which may limit problem behavior), and (iv) orientation toward success (i.e., consumption of illicit drugs reduces the physical capacities and therefore alters the sport results and achievement). On the other hand, authors also recognized sport as a social activity that may present a certain risk of a higher likelihood of substance misuse, including cannabis consumption. Putting it altogether, many but not all sport characteristics may reduce the risk of cannabis use [18]. Supportively, studies which investigated the associations between sport factors and illicit drug misuse reported inconsistent findings [19-21].” (Please see 4th paragraph of the Introduction)
On p.134 – what value is considered “appropriate”?
RESPONSE: According to cited literature, the IC < 0.10 is considered as “small enough”. It is now stated. Thank you
It is not clear how the familial SES questions were brought together
RESPONSE: Thank you for noticing it. These three questions were chosen as determinant of SES based on previous studies which specifically investigated the SES as potential correlate of substance misuse. It is now clearly stated. Text reads: “The familial social status was determined by three questions: one asking for self-determined socioeconomic status (SES; under average, average, and above average), one question on maternal education and one question for paternal-education (three possible answers for both maternal and paternal education: elementary school, high school, and college/university degree). Although these three variables theoretically determine different social facets (i.e. economics, educational status), similar indices are used previously in order to identify construct of social status [33].” (Please see 2nd paragraph of the Instrument subsection).
Sex and urban rural status should be included in al models as covariates. These are important covariates and there is no theoretical or analytical reason not to. Thereby, only model 3 is needed and the coefficients for gender and urban/rural ought to be reported on.
RESPONSE: As you suggested in this version of the manuscript only logistic regression models adjusted for sex and environment. Therefore, statistics, tables and corresponding part of text are amended accordingly. For Statistics it reads: “Since preliminary statistical procedures showed significant association between: (i) gender and cannabis misuse, and (ii) urban/rural environment and cannabis misuse (see later text for details on differences between genders and urban/rural environment), logistic regressions were adjusted for gender and urban/rural environment (Model 3).” Thank you!
The tables should include an n
RESPONSE: The number of observations is included in tables (First column in each table). Thank you
On line 229 – it is not clear how it was determined that the trajectory was significantly increased
RESPONSE: Indeed, we did not specify the results of the Chi square for differences between prevalence at baseline and follow-up. It is amended. Text reads: “The prevalence of cannabis misuse increased from 28% at study baseline to 37% at follow-up (χ2: 12.59, p < 0.01), with the similar increase across genders.“ (Please see begining of the Results section) Thank you!
On lines 241-242 – confidence intervals should be given on the estimates to make better comparisons
RESPONSE: The 95%CI is included. Text reads: “According to the presented results, we can say that the prevalence of cannabis consumption in our sample (approximately 30% users; 95%CI: 26-33%) is higher than the average value reported for the whole territory of Croatia (22% of users; 95%CI: 20-24%) [37].” (Please see beginning of subsection 4.1. Prevalence and trajectory). Thank you.
Line 251- “unfortunately” is values based
RESPONSE: Amended accordingly. The term “unfortunately” is replaced with “but”. Text reads: “The high prevalence of cannabis consumption in our sample is in accordance with public health guidelines obtained in a recent ESPAD report. In short, from 1995-2003, an almost triple growth of illicit drug use among youth (age 15-16 years) was noted in Croatia, with a slight decrease in 2007 and 2011, but the values increased again in 2015 to the 2003 level [38].” (Please see 2nd paragraph of the subsection 4.1. Prevalence and trajectory): Thank you.
Lines 166-167 are not necessary
RESPONSE: Amended accordingly (deleted)
On line 196 it seems that model 1 adjusts for no covariates but this is not the case in the tables
RESPONSE: As you specified in one of the previous comments, the Model 1 and Model 2 are not even presented in the revised version of the article. Therefore this part of the text is not included in the revised version.
Some of the discussion is over-reaching the data- i.e., line 265, line 287-288
RESPONSE: Amended accordingly (deleted)
The final paragraph is vague.
RESPONSE: The paragraph is systematically rewritten and now reads: “Although further investigations are needed to support previous considerations about negative influence of sport factors on cannabis initiation, previous discussion can be indirectly supported by results of some previous studies where authors investigated correlated sport factors with consumption of illicit drugs among adolescents. For example, a very recent longitudinal study of almost 2000 Canadian adolescent females who participated in team sports revealed an increase in the odds of becoming more frequent cannabis users [51]. Additionally, our results about increased risk for cannabis initiation in athletic adolescents is consistent with some cross-sectional and prospective reports in the territory of southeastern Europe [20,35]. Collectively, results highlight the possibility that under some specific circumstances, sport-participation should be observed as being negatively related to consumption of illicit drugs (including cannabis).” (Please see final paragraph of the Discussion prior Study Limitations. Thank you.)
Minor issues:
In the abstract on line 18, remove “(predictors)”
RESPONSE: Removed
On line 27 – “was risk” should read “was a risk”
RESPONSE: Corrected.
Line 33- the statement that the cannabis use found in the study is “alarming” is quite subjective. It would be more informative to compare the prevalence to other regions or remove this statement.
RESPONSE: Statement is removed as suggested.
On line 129 – add who after adolescents
RESPONSE: Amended accordingly.